# Identifying hopelessness in population research: a validation study of two brief measures of hopelessness

Lindsay Fraser,[1] Matthew Burnell,[1] Laura Currin Salter,[2]
Evangelia-Ourania Fourkala,[1] Jatinderpal Kalsi,[1] Andy Ryan,[1]
Sue Gessler,[1,2] Yori Gidron,[3] Andrew Steptoe,[4] Usha Menon[1]

▶ Prepublication history and additional material is available. To view please visit the journal (http://dx.doi.org/10.1136/bmjopen-2014-005093).

For numbered affiliations see end of article.

**Correspondence to**
Lindsay Fraser;
l.fraser@ucl.ac.uk

## ABSTRACT

**Objective:** Hopelessness is an important construct in psychosocial epidemiology, but there is great pressure on the length of questionnaire measures in large-scale population and clinical studies. We examined the validity and test–retest reliability of two brief measures of hopelessness, an existing negatively worded two-item measure of hopelessness (Brief-H-Neg) and a positively worded version of the same instrument (Brief-H-Pos).

**Design:** Cohort study.

**Setting:** Control arm of the UK Collaborative Trial of Ovarian Cancer Screening.

**Participants:** A non-clinical research-based sample of 5000 postmenopausal women selected from 56 512 participants.

**Primary and secondary outcome measures:** Spearman's rank correlation of brief measures of hopelessness with the Beck Hopelessness Scale (BHS). Spearman's rank correlation with the Centre for Epidemiological Studies Depression Scale (CES-D) and change in mean score on repeat testing.

**Methods:** Two short hopelessness measures, a negatively worded brief measure of hopelessness (Brief-H-Neg) and a positively worded brief measure of hopelessness (Brief-H-Pos), were administered by postal questionnaire to 5000 women together with the 20-item BHS and 20-item CES-D. The Brief-H-Neg and Brief-H-Pos were readministered to 500 women after a 2-week interval.

**Results:** 2413 postmenopausal women (mean age 68.9 years) completed the questionnaire. The Brief-H-Neg and Brief-H-Pos correlated 0.93 and 0.87 with the BHS after correction for attenuation and their association with the CES-D mirrored that seen with the BHS (Spearman's rank correlation 0.88 and 0.68, respectively). There was no change in mean scores on the two measures with repeat testing in the 433 women who completed them and test–retest reliability was good (intraclass correlations Brief-H-Neg 0.67 and Brief-H-Pos 0.72).

**Conclusions:** These findings provide support for the validity of the Brief-H-Neg and Brief-H-Pos. These brief measures are likely to be useful in large population studies assessing hopelessness.

**Trial registration number:** NCT00058032.

## Strengths and limitations of this study

- The strength of this study is the large sample size.
- Limitations include generalisability of the results beyond older women and the modest response rate.
- It is not known whether the positively phrased measure of hopelessness is associated with less participant distress compared to the negatively phrased measure.

## INTRODUCTION

Hopelessness is the subjective appraisal of negative expectations about the occurrence of highly valued outcomes coupled with the sense that one lacks control over desired events in the future.[1] Hopelessness has been related to the onset and prognosis of mental and physical health outcomes including the development of depression,[1] suicidal ideation,[2] hypertension,[3] subclinical atherosclerosis,[4][5] adaptation following acute cardiac events[6] and progression of carotid atherosclerosis.[4] In the psycho-oncology literature, hopelessness has been found to predict prognosis in various cancers including breast and haematological cancers,[7][8] although the evidence is not consistent.

Hopelessness has been measured in clinical and population research in a variety of ways including systematic interviews[9] and validated psychometric measures such as the Beck Hopelessness Scale (BHS)[10] and the Mental Adjustment to Cancer scale.[11] There is great pressure in large-scale population studies on questionnaire size due to the volume of clinical and demographic variables that must be collected. Everson *et al*[12] devised a two-item measure of hopelessness which has been used in a number of cardiovascular studies.[3][4][12] The reliability of this instrument and its relationship with standard measures has not been established. An

additional issue concerns the negative valence of the items (eg, "The future seems to me to be hopeless and I can't believe that things are changing for the better"). In preliminary work for the large study in which this research is embedded, some respondents found these items upsetting and this has been confirmed by others.[13] We devised a positively worded two-item version. We compared both brief measures with established measures of hopelessness and depressive symptoms in a large population sample, and assessed their reliability.

## METHODS
### Participants
Five thousand participants were selected from 56 512 postmenopausal women in the control arm of the UK Collaborative Trial of Ovarian Cancer Screening (UKCTOCS,[14] ISRCTN22488978). The mean age of women invited was 69.6±6.1 years (range 57–85).

### Procedure
A postal questionnaire comprised of measures of hopelessness and depression was sent to 5000 women (Time 1, T1). After a 2-week interval (Time 2, T2), 500 respondents were asked to repeat the Brief-H-Neg (n=250) or the Brief-H-Pos (n=250) to assess test–retest reliability. Selection of the retest cohort was staggered based on the date of T1 questionnaire return, as early and late responders may differ on levels of hopelessness or depression.[15]

### Measures
The Brief-H-Neg is a two-item measure of hopelessness comprised of negatively valenced statements: "The future seems to me to be hopeless and I can't believe that things are changing for the better"; "I feel that it is impossible to reach the goals I would like to strive for."[12] Everson *et al* selected these from a battery of psychosocial measures used in the Kuopio Ischemic Heart Disease study, defining hopelessness as negative expectancies about oneself and the future. Respondents indicate agreement on a five-point scale (range 2–10), higher scores indicate higher hopelessness (see online supplementary appendix A).

The Brief-H-Pos was derived by reversing the tone of the Brief-H-Neg statements from negative to positive and reverse scoring: "The future seems to me to be hopeful and I believe that things are changing for the better"; "I feel that it is possible to reach the goals I would like to strive for" (see online supplementary appendix B).

The BHS is a validated 20-item true–false measure assessing current levels of hopelessness.[10] Items include pessimistic statements ("There's no use in really trying to get something I want because I probably won't get it") and optimistic ones ("I look forward to the future with hope and enthusiasm"). Pessimistic ratings are summed (range 0–20); higher scores indicate higher hopelessness.

The Centre for Epidemiological Studies Depression Scale (CES-D) is a validated 20-item measure of depressive symptoms.[16] Responses are based on the frequency of occurrence during the past week using a four-point scale (range 0–60); higher scores indicate more frequent symptoms of depression.

### Analyses
Internal consistency was based on coefficient $\alpha$[17] with $\alpha$ cut-off points 0.70–0.79 described as *adequate* and ≥0.80 as *high*.[18] Stability was evaluated using test–retest reliability based on the intraclass correlation coefficient (ICC) with cut-offs ≤0.40 for *poor*, 0.41–0.59 *fair*, 0.60–0.74 *good*, ≥0.75 *excellent*.[19] The estimated variance components derived from a one-way random effects model were used to calculate ICCs.[20] The relationship between study measures was assessed using Spearman's rank correlations (CIs were estimated using bootstrapping with 1000 iterations).[21] To estimate the strength of correlations between study measures, a correction for attenuation arising from measurement error was applied: $\rho_{xy} = r_{xy}/\sqrt{(r_{xx}.r_{yy})}$,[22] [23] where $\rho_{xy}$=true correlation between x and y, $r_{xy}$=observed correlation between x and y, $r_{xx}$=estimated reliability of x and $r_{yy}$=estimated reliability of y. We used published test–retest reliability estimates for $r_{xx}$ and $r_{yy}$: BHS 0.69[24] and CES-D 0.67.[16] In the absence of published test–retest data for the Brief-H-Neg/Brief-H-Pos, we used the ICCs reported in this study. Data were analysed using STATA V.12.1.

## RESULTS
### Sample characteristics
The questionnaire was returned by 2413 women (48.3%; T1; table 1). Respondents reported significantly higher

**Table 1** Description of respondents' characteristics

| | Respondents (N=2413) |
|---|---|
| Age in years (mean±SD) | 68.9±5.9 (range 57–84) |
| Ethnicity n (%) | |
| White | 2376 (98.7) |
| Black | 11 (0.5) |
| Asian | 7 (0.3) |
| Other | 14 (0.6) |
| Unknown | 5 (0.2) |
| Education n (%) | |
| Higher (university, professional) | 819 (33.9) |
| Some (O' level, A' level, clerical) | 955 (39.6) |
| None | 610 (25.3) |
| Unknown | 29 (1.2) |
| Hopelessness (mean±SD) | |
| Brief-H-Neg | 4.42±2.21 (n=2402) |
| Brief-H-Pos | 4.74±1.85 (n=2393) |
| BHS | 4.81±4.49 (n=2400) |
| Depression (mean±SD) | |
| CES-D | 12.44±10.39 (n=2395) |

BHS, Beck Hopelessness Scale; CES-D, Centre for Epidemiological Studies Depression Scale.

levels of education than non-respondents, were younger and more likely to be Caucasian (differences were not clinically significant, due to their small magnitude). One hundred and fifteen respondents (4.77%) scored CES-D ≥16/60, a cut-off indicative of clinically significant depressive symptomatology, suggesting that this cohort is not unusually depressed.

## Concurrent validity
The Brief-H-Neg and Brief-H-Pos measures correlated well with the BHS and mirrored the positive association seen between the BHS and the CES-D (table 2).

## Stability
In total, 433/497 (87.1%) women completed the Brief-H-Neg (n=221) or Brief-H-Pos (n=212) on two occasions. Brief-H-Neg, T1 M=4.64±1.74 (n=248), T2 M=4.29±2.39 (n=221); Brief-H-Pos, T1 M=4.61±1.878 (n=249), T2 M=4.57±1.96 (n=212). The short-term test–retest reliability of both measures was good: Brief-H-Neg ICC=0.67 (95% CI 3.98 to 4.49) and Brief-H-Pos ICC=0.72 (95% CI 4.39 to 4.83).

## Reliability
All study measures demonstrated good internal consistency: Brief-H-Neg α 0.80, Brief-H-Pos α 0.77, BHS α 0.89, CES-D α 0.90. α for the Brief-H-Neg and Brief-H-Pos was lower than the longer BHS and CES-D (α is known to rise as the number of items increases).

## DISCUSSION
A brief measure is needed to examine the role of hopelessness on mental and physical health outcomes in large population studies. We examined the validity and reliability of two brief measures of hopelessness in a large non-clinical sample, one negatively valenced (Brief-H-Neg) and one positively valenced (Brief-H-Pos). Both were shown to correlate strongly with the longer BHS and mirror the positive correlation seen between the BHS and a measure of depression, providing evidence of concurrent validity, with adequate internal consistency and test–retest reliability.

The sizes of the 2-week retest correlations for the brief measures reported in our non-clinical sample (0.67 and 0.72) are similar to those reported for the BHS in a sample of university undergraduates over a 3-week retest interval (0.67, female students) or a 10-week interval

(0.75).[25] [26] Studies assessing the retest reliability of hopelessness instruments have reported varying retest intervals. Hopelessness may be conceptualised as a temporary mood state reflecting a person's response to challenging circumstances, or a more enduring trait reflecting a habitual outlook on many aspects of life.[27] Most commonly used measures of hopelessness, including the BHS, do not distinguish between state and trait hopelessness. If hopelessness is an enduring trait, measures of hopelessness would be expected to have high test–retest reliability. A measure that does address the state versus trait distinction, the State-Trait Hopelessness Scale, has reported retest correlations of state and trait hopelessness over a 6-week interval (state 0.65, trait 0.74) and over a 6-month interval (state 0.61, trait 0.78) in hospitalised patients with coronary heart disease.[28] Again, the sizes of these retest correlations are not dissimilar to those seen in the brief measures reported in our study after a 2-week interval.

The selection of a measure is determined to an extent by the practical context of the investigation. Very brief measures necessarily sacrifice some level of detail compared with their longer counterparts.[29] A pooled analysis and meta-analysis of 22 studies involving ultrashort (one-item, two-item, three-item or four-item) tests concluded that two-item and three-item measures of depression identify 8 out of 10 cases in primary care settings, albeit at the expense of a high false-positive rate.[30] This makes them inappropriate diagnostic tests for clinical decision-making, but suitable as screening tools in primary care as well as in population cohort research where participants have to complete a number of demographic and clinical questions in addition to psychological measures.[31]

Our data suggest that while 2-item measures of hopelessness may not have the detail of the 20-item BHS measure, they do have adequate reliability to be used in large population-based studies. The reduced burden on participants may encourage a high response rate. The five-point Likert response scales of the Brief-H-Neg and Brief-H-Pos provide a reasonable range of scores to work with. However, if information on the hypothesised affective, motivational and cognitive aspects of hopelessness is required in order, for example, to target a therapeutic intervention, the 20-item BHS would be more suitable, because a total score for each dimension can be derived from the summed individual items of the scale.[10]

The results of this study provide preliminary support for the construct validity of both brief measures of hopelessness but further testing of their construct validity is required, along with tests of their predictive validity on physical and mental health outcomes. It would be helpful to examine the psychometric properties of both brief measures in a psychiatric sample where higher levels of hopelessness are expected, such as a group of hospitalised patients who have attempted suicide.[32] There is good evidence that hopelessness is associated with suicidal ideation and is recognised as a better

**Table 2** Correlation between measures of hopelessness and depression

|       | Brief-H-Neg (n) | Brief-H-Pos (n) | BHS (n)     |
|-------|-----------------|-----------------|-------------|
| BHS   | 0.93 (2393)     | 0.87 (2384)     |             |
| CES-D | 0.88 (2379)     | 0.68 (2392)     | 0.87 (2379) |

BHS, Beck Hopelessness Scale; CES-D, Centre for Epidemiological Studies Depression Scale.

predictor for suicidal intent than depression.[33] Moreover, brief measures of hopelessness derived from the BHS, including a four-item scale and, to a lesser extent, a single item, have been shown to perform as well as the 20-item BHS in identifying people with suicidal ideation.[34] The predictive validity for the Brief-H-Neg on physical health outcomes has been shown in studies exploring the relationship between hopelessness and disease incidence and mortality, and this remains to be addressed for the Brief-H-Pos.[3–5 12]

There are some limitations to this study. First, the sample of older women limits the generalisability of the results. It would be useful to validate the Brief-H-Neg and Brief-H-Pos in a general population sample and to generate normative data, as has been shown for two-item measures of depression (Patient Health Questionnaire, PHQ-2) and anxiety (Generalized Anxiety Disorder Scale, GAD-2).[35 36] Second, the response rate of 48.3% is modest, although importantly there was no evidence of bias between responders and non-responders and the sample of responders is large. It is perhaps unsurprising that many of the women invited from the control arm of an Ovarian Cancer Screening study were not motivated to take part in this nested study assessing brief measures of hopelessness. Finally, we did not directly test the assumption that those suffering from low mood may find it difficult to be confronted with the negatively phrased questions of the Brief-H-Neg compared with the positively phrased Brief-H-Pos.

## CONCLUSION

Everson *et al*'s negatively valenced measure of hopelessness (Brief-H-Neg) and the positively valenced measure (Brief-H-Pos) [which was] developed as a potentially less stressful measure in health research, have been shown to be valid and reliable measures of hopelessness. Further testing to verify their construct validity is warranted. Meanwhile, the findings suggest that these brief measures are fit for purpose in large-scale population studies investigating the association of hopelessness and health outcomes. Evidence of a consistent association with mortality in such studies would add impetus to the search for interventions that can modify the risk.

**Author affiliations**
[1]Department of Women's Cancer, UCL Institute for Women's Health and NIHR University College London Hospitals Biomedical Research Centre, London, UK
[2]University College London Hospitals NHS Foundation Trust, London, UK
[3]Faculty of Medicine & Pharmacy, Vrije Universiteit Brussel (VUB), Brussels, Belgium
[4]Free University of Brussels (VUB), Faculty of Medicine & Pharmacy, Brussels, Belgium

**Contributors** All authors contributed to the study design. JK and AR collected the data. MB conducted the data analysis. LF, MB, UM and AS interpreted the data. LF, UM and AS drafted the manuscript. All authors revised and approved the final version of the manuscript.

**Funding** UKCTOCS is core funded by the Medical Research Council (G0801228), Cancer Research UK (C1479/A2884), The Department of Health,

with additional support from The Eve Appeal, Special Trustees of Barts and the London, Special Trustees of UCLH.

**Competing interests** None.

**Ethics approval** Ethical approval was obtained from the NRES Committee North West-Haydock (00/8/034).

**Provenance and peer review** Not commissioned; externally peer reviewed.

**Data sharing statement** The data from the current study are available to specific researchers at the Gynaecological Cancer Research Centre, UCL Institute for Women's Health.

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
