## [Reviewer comments · BMJ Open]

Some articles will have been accepted based in part or entirely on reviews undertaken for other BMJ Group journals. These will be reproduced where possible.

ARTICLE DETAILS

TITLE (PROVISIONAL)	Identifying hopelessness in population research: a validation study of two brief measures of hopelessness.
AUTHORS	Fraser, Lindsay; Burnell, Matthew; Salter, Laura; Fourkala, Evangelia-Ourania; Kalsi, Jatinderpal; Ryan, Andy; Gessler, Sue; Gidron, Yori; Steptoe, Andrew; Menon, Usha

VERSION 1 - REVIEW

REVIEWER	James Overholser Professor of Psychology Case Western Reserve University Cleveland Ohio USA
REVIEW RETURNED	20-Mar-2014

GENERAL COMMENTS	The authors present a research study that examines the assessment of hopelessness in a sample of 2,413 non-psychiatric adult females. The study examines some important clinical issues, and the text is well written. However, I have a few concerns about the study. The study focuses on the development of two new brief measures of hopelessness. The Brief Hopelessness Negative scale includes 2 items, and the Brief Hopelessness Positive scale includes 2 items. These items are scored from 1-5, and summed across the 2 items, for scores that can range from 2-10. I usually find brief measures to be useful in clinical research, but I worry that these scales are too brief. Furthermore, i am not sure if coefficient alpha is an appropriate measure of internal consistency when a scale includes only two items. The authors report correlations across measures. It seems premature to discuss construct validity when this is really a preliminary study on scale development. Concurrent validity may be the more accurate term. Among the sample, 212 women completed the positive hopelessness scale two weeks later, and 221 women completed the negative hopelessness scale two weeks after the initial assessment. These sub-samples do not overlap. Retest correlations were modest (.67 and .72), which seem rather low for a two-week retest interval, especially if hopelessness is viewed as a cognitive perspective distinct from temporary mood states. I would have preferred to see a longer retest interval. More importantly, it would seem helpful to examine the psychometric properties of these scales in a psychiatric sample where elevated scores might be expected to occur. Finally, I wonder if there would be benefits of combining these two brief scales into one 4-item measure, which would still be quite short and easy to administer.
---

REVIEWER	Maurizio Pompili, M.D., Ph.D., Professor of Suicidology Sapienza University of Roma, Italy
REVIEW RETURNED	31-Mar-2014

GENERAL COMMENTS	The authors reported interesting results of their investigation searching for the validation of two brief measures of hopelessness in a cohort of post-menopausal women. The paper is of interest for the journal. I have minor shortcomings to mention. First, although the objective is clear, the two brief measures of hopelessness may be not so familiar to readership. I therefore suggest to add some information such items or something else in an appendix. This is because the BHS is well-know and considered a comparative measure. Second, limitations are not properly reported; authors should report limitations for this study. The conclusions need some improvement with more focus on their investigation and implications for future perspectives. Given the fact that hopelessness has been reported as a proxy for suicide risk I suggest to mention such implication and cite the following paper: Exploring the phenomenology of suicide. Suicide and Life-Threatening Behavior 2010
---

VERSION 1 – AUTHOR RESPONSE

Reviewer 1: Prof James Overholser

1. "I usually find brief measures to be useful in clinical research, but I worry that these scales are too brief."

We agree with the reviewer that very brief instruments necessarily sacrifice some level of detail, compared to longer measures and have expanded our discussion on this aspect to address his concern and to describe appropriate use of very brief measures (p.9, line 30).

"The selection of a measure is determined to an extent by the practical context of the investigation. Very brief measures necessarily sacrifice some level of detail compared with their longer counterparts.[29] A pooled analysis and meta-analysis of 22 studies involving ultra-short (one-, two-, three- or four-item) tests concluded that 2-item and 3-item measures of depression identify 8 out of 10 cases in primary care settings, albeit at the expense of a high false positive rate.[30] This makes them inappropriate diagnostic tests for clinical decision making, but suitable as screening tools in primary care as well as in population cohort research where participants have to complete a number of demographic and clinical questions in addition to psychological measures.[31]

Our data suggest that while 2-item measures of hopelessness may not have the detail of the 20-item BHS measure, they do have adequate reliability to be used in large population based studies. The reduced burden on participants may encourage a high response rate. The 5-point Likert response scales of the Brief-H-Neg and Brief-H-Pos provide a reasonable range of scores to work with. However, if information on the hypothesised affective, motivational and cognitive aspects of hopelessness is required in order for example to target a therapeutic intervention, the 20-item BHS would be more suitable, because a total score for each dimension can be derived from the summed individual items of the scale.[10]"

2. "I am not sure if coefficient alpha is an appropriate measure of internal consistency when a scale includes only two items."

Coefficient alpha has been used to assess the internal consistency of other 2-item measures of psychological constructs, including the PHQ-2 depression scale and GAD-2 anxiety scale (Löwe, Wahl et al. 2010). We believe that coefficient alpha is appropriate on a 2-item scale to assess internal consistency, as long as the scale is one-dimensional and the measures are 'tau-equivalent'. As the number of items in a measure increases, these assumptions become less tenable (Aish 2000, Aish, Wasserman et al. 2001). In our study analysis, we examined the internal consistency of the 2-item Brief-H-Neg and Brief-H-Pos scales using coefficient alpha and Spearman-Brown half-split reliability. Both methods gave very similar results: 0.77 for the Brief-H-Neg and 0.80 for the Brief-H-Pos.

3. "The authors report correlations across measures. It seems premature to discuss construct validity when this is really a preliminary study on scale development. Concurrent validity may be the more accurate term."

We thank Prof Overholser for drawing our attention to this. We agree that construct validity cannot be demonstrated in a single study, but is an on-going process of evaluation and refinement over the course of many studies. We have made changes in the manuscript to reflect this. The text referring to 'construct validity' has been changed to 'concurrent validity' in the Results (p.7, line 46) and Discussion (p.8, line 52).

4. "Retest correlations were modest (.67 and .72), which seem rather low for a two-week retest interval, especially if hopelessness is viewed as a cognitive perspective distinct from temporary mood states. I would have preferred to see a longer retest interval."

The size of the 2-week retest correlations for the brief measures reported in our non-clinical sample (0.67 and 0.72) are similar to those reported for the BHS in a sample of university undergraduates over a 3-week retest interval (0.67, female students) or a 10-week interval (0.75) (Holden and Fekken 1988, Fisher and Overholser 2013). We have now added this to the discussion (p.8, line 56). We agree with the reviewer on the relevance of the retest interval when evaluating the reliability of a measure of hopelessness and have added a section in the manuscript addressing this (p.9, line 7-28). "The size of the 2-week retest correlations for the brief measures reported in our non-clinical sample (0.67 and 0.72) are similar to those reported for the BHS in a sample of university undergraduates over a 3-week retest interval (0.67, female students) or a 10-week interval (0.75).[25, 26] Studies assessing retest reliability of hopelessness instruments have reported varying retest intervals. Hopelessness may be conceptualised as a temporary mood state reflecting a person's response to challenging circumstances, or a more enduring trait reflecting a habitual outlook on many aspects of life.[27] Most commonly used measures of hopelessness, including the BHS, do not distinguish between state and trait hopelessness. If hopelessness is an enduring trait, measures of hopelessness would be expected to have high test-retest reliability. A measure that does address the state versus trait distinction, the State-Trait Hopelessness Scale, has reported retest correlations of state and trait hopelessness over a 6-week interval (state 0.65, trait 0.74) and over a 6-month interval (state 0.61, trait 0.78) in hospitalised patients with coronary heart disease.[28] Again, the size of these retest correlations are not dissimilar to those seen in the brief measures reported in our study after a 2-week interval."

Other hopelessness instruments have reported lower retest correlations after a 3-week interval using the Helplessness, Hopelessness, and Helplessness Scale in outpatients diagnosed with anxiety disorders (0.52), and after a 6-week interval using the Hopelessness Scale for Children in psychiatric child inpatients (0.52) (Kazdin, Rodgers et al. 1986, Vatan 2011).

5. "It would seem helpful to examine the psychometric properties of these scales in a psychiatric sample where elevated scores might be expected to occur."

We have included a section in the Discussion (p.10, line 13), suggesting that the 2 brief measures are evaluated in settings where higher levels of hopelessness are expected, e.g., a group of hospitalised patients who have attempted suicide, as there is good evidence that hopelessness is associated with suicidal ideation.

“It would be helpful to examine the psychometric properties of both brief measures in a psychiatric sample where higher levels of hopelessness are expected, such as a group of hospitalised patients who have attempted suicide.[32]”

6. “I wonder if there would be benefits of combining these two brief scales into one 4-item measure, which would still be quite short and easy to administer.”

We do not think there would be a benefit in combining the four statements from these two brief measures into one because: (1) A key aim of our study was to assess the validity of a brief positively worded measure of hopelessness (the Brief-H-Pos), which may be useful in minimising participant distress, and this precludes the inclusion of negatively worded statements in the same measure. (2) A 4-item hopelessness scale derived from the BHS with two positively worded statements (items 6 ‘In the future I expect to succeed in what concerns me most’ and 15 ‘I have great faith in the future’) and two negatively worded (items 7 ‘My future seems dark to me’ and 9 ‘I just don’t get the breaks and there is no reason to believe I will in the future’), is already available for researchers who wish to use a slightly longer measure with a mixed affective tone (Aish 2000, Yip and Cheung 2006).

Reviewer 2: Prof Maurizio Pompili

1. The two brief measures of hopelessness may be not so familiar to readership. I therefore suggest to add some information such items or something else in an appendix.

While the full wording of the 2 statements that make up the Brief-H-Neg and Brief-H-Pos measures are included in the Methods section of the manuscript, we have added appendices showing the two measures as used in our study questionnaire and the scoring instructions (p.5, lines 36 and 44). We think this will help readers appreciate the brevity and simplicity of the measures.

2. Authors should report limitations for this study.

We have added a paragraph in the Discussion on the limitations of the study (p.10 line 36).

“There are some limitations to this study. Firstly, the sample of older women limits the generalizability of the results. It would be useful to validate the Brief-H-Neg and Brief-H-Pos in a general population sample and to generate normative data, as has been shown for 2-item measures of depression (PHQ-2) and anxiety (GAD-2).[35] Secondly, the response rate of 48.3% is modest, although importantly there was no evidence of bias between responders and non-responders and the sample of responders is large. It is perhaps unsurprising that many of the women invited from the control arm of an ovarian cancer screening study were not motivated to take part in this nested study assessing brief measures of hopelessness. Lastly, we did not directly test the assumption that those suffering from low mood may find it difficult to be confronted with the negatively phrased questions of the Brief-H-Neg compared with the positively phrased Brief-H-Pos.”

3. “The conclusions need some improvement with more focus on their investigation and implications for future perspectives.”

We have addressed the reviewer’s comments in the Conclusion (p. 11, line 5).

“Both Everson et al’s negatively valenced measure of hopelessness (Brief-H-Neg) and the positively

valenced measure (Brief-H-Pos) developed as a potentially less stressful measure for participants in health research have been shown to be valid and reliable measures of hopelessness. Further testing to verify their construct validity is warranted. Meanwhile the findings suggest that these brief measures are fit for purpose in large scale population studies investigating the association of hopelessness and health outcomes. Evidence of a consistent association with mortality in such studies would add impetus to the search for interventions that can modify the risk.”

4. “Given the fact that hopelessness has been reported as a proxy for suicide risk I suggest to mention such implication and cite the following paper: Exploring the phenomenology of suicide. Suicide and Life-Threatening Behavior 2010.”

The association between hopelessness and risk of suicide has been included in the Introduction (p. 4, line 12) and the Discussion (p.10, line 21), along with Prof Pompili’s suggested reference (Pompili 2010).

References

Aish, A.-M., D. Wasserman and E. Renberg (2001). Does Beck's Hopelessness Scale really measure several components? *Psychological medicine* 31(2): 367-372.

Aish, A. M. W., D. (2000). The Dimensionality of Beck's Hopelessness Scale. Report 3. Centre for Suicide Research and Prevention. Stockholm, Karolinska Institute.

Fisher, L. B. and J. C. Overholser (2013). Refining the Assessment of Hopelessness: An Improved Way to Look to the Future. *Death Studies* 37(3): 212-227.

Holden, R. R. and G. C. Fekken (1988). Test-retest reliability of the hopelessness scale and its items in a university population. *Journal of Clinical Psychology* 44(1): 40-43.

Kazdin, A. E., A. Rodgers and D. Colbus (1986). The Hopelessness Scale for Children: Psychometric characteristics and concurrent validity. *Journal of consulting and clinical psychology* 54(2): 241.

Löwe, B., I. Wahl, M. Rose, C. Spitzer, H. Glaesmer, K. Wingenfeld, A. Schneider and E. Brähler (2010). A 4-item measure of depression and anxiety: validation and standardization of the Patient Health Questionnaire-4 (PHQ-4) in the general population. *Journal of affective disorders* 122(1): 86-95.

Pompili, M. (2010). Exploring the phenomenology of suicide. *Suicide and Life-Threatening Behavior* 40(3): 234-244.

Vatan, S. E., S.; Lester, D. (2011). Test-retest reliability and construct validity of the Helplessness, Hopelessness, and Haplessness Scale in patients with anxiety disorders 1. *Psychological reports* 108(2): 673-674.

Yip, P. S. and Y. B. Cheung (2006). Quick assessment of hopelessness: a cross-sectional study. *Health and quality of life outcomes* 4(1): 13.

VERSION 2 – REVIEW

REVIEWER	Maurizio Pompili, M.D., Ph.D. Dept. of Neurosciences, Mental Health and Sensory Organs. Director, Suicide Prevention Center, Sant'Andrea Hospital, Sapienza University of Rome, Italy
REVIEW RETURNED	04-May-2014
GENERAL COMMENTS	The authors addressed my comments and the paper appears suitable for possible publication in the journal.